# Finetuning the Geospatial Foundation model for Land Cover mapping

Devyani Lambhate
Ayush jain
Kamal Das
Ranjini Bangalore
devyani.lambhate1@ibm.com
ayush.jain@ibm.com
kdas3@in.ibm.com
rangurup@in.ibm.com
IBM Research
India

Johannes Jakubik
Johannes.Jakubik1@ibm.com
IBM Research
Switzerland

Bianca Zadrozny
biancaz@br.ibm.com
IBM Research
Brazil

## ABSTRACT

Land use and land cover (LULC) play pivotal roles in achieving several sustainable development goals established by United Nations member states for the social, economic, and environmental advancement of our planet and its inhabitants. Understanding LULC and its dynamics is crucial for gaining insights into the changing composition and spatial distribution of land surface features across diverse landscapes. While researchers have explored various AI/ML approaches using remote sensing images spanning several decades, existing LULC mapping techniques encounter challenges related to accuracy, the need for substantial labeled data for training, and adaptability to different geographical regions, among others. Recent advances in foundational models have gained significant traction due to their ability to alleviate labeled data scarcity issues. Therefore, in this study, we propose a fine-tuning strategy utilizing a cutting-edge geospatial foundation model jointly developed by IBM and NASA, known as Prithvi [17], to address existing challenges such as lesser requirements of labeled data and accuracy. This paper presents the results achieved using Prithvi for LULC mapping with relatively small training dataset (565 total $224 \times 224$ pixel$^2$ images). We conduct a performance comparison of the Prithvi model with traditional deep learning-based U-Net model and a large foundational model known as Vision Transformer (ViT). The results demonstrate that Prithvi surpasses U-Net and ViT in terms of mean Intersection over Union (IoU) across several LULC classes.

## KEYWORDS

Land Use Land Cover, LULC, Remote Sensing, Foundational model

**Unpublished working draft. Not for distribution.**

ACM Reference Format:
Devyani Lambhate, Ayush jain, Kamal Das, Ranjini Bangalore, Johannes Jakubik, and Bianca Zadrozny. 2018. Finetuning the Geospatial Foundation model for Land Cover mapping. In *Proceedings of Make sure to enter the correct conference title from your rights confirmation emai (Conference acronym 'XX)*. ACM, New York, NY, USA, 7 pages. https://doi.org/XXXXXXX.XXXXXXX

## 1 INTRODUCTION

The 17 sustainable development goals (SDGs)[1] form the core of the agenda for sustainable development adopted by all the United Nations Member States in 2015 [12]. The SDGs are an all-encompassing guide aimed at achieving social, economic and environmental conservation, preservation and development of the planet and its inhabitants in a sustainable manner. Eight of the SDGs, namely, (1) zero hunger, (2) clean water and sanitation, (3) affordable and clean energy, (4) industry, innovation and infrastructure, (5) sustainable cities and communities, (6) responsible consumption and production, (7) climate action, (8) life on land are closely linked to how land is used, and what covers the land. LULC information which enables identification and quantification of land covered by vegetation, natural features and man-made structures is required to devise strategies to implement the above eight SDGs and also measure the efficacy of the devised strategies. Key stakeholders, including governments, policy makers, private enterprises, and individuals, therefore rely on LULC and its spatio-temporal evolution to balance the needs of society and the environment. Hence it is critical to generate accurate global LULC information at spatio-temporal scales required to achieve the above SDGs.

Currently, LULC maps are generated through a combination of ground truth data, remote sensing, and AI techniques. These maps vary in spatial resolution, ranging from 5 meters [29] to 500 meters [10], and are typically updated on an annual basis. However, state-of-the-art LULC generation methods face several limitations: (1) They require a large amount of labeled data, whereas in reality, data scarcity is a significant issue [8]. (2) They struggle with generalizability across different land cover classes. (3) Scaling these methods is challenging and often incurs significant costs.

[1] https://sdgs.un.org/goals

In this paper, we aim to investigate the adaptability of foundation models in addressing some of above challenges while generating LULC maps. Specifically, we aim to demonstrate their capability to achieve comparable or superior results (with respect to state of art techniques) with fewer ground truth data points and generalization across land classes, thereby addressing the challenges of limited ground truth data and diverse land classes.

The rest of the paper is organised as follows. Section 2 briefly describes the related work. The Prithvi model and its adaptation for LULC mapping is described in section 3. Details of data used for pre-training and fine-tuning are given in section 4. The pre-trained Prithvi model and approach for fine-tuning it is described in 5. The experiments carried out are laid out in section 6. The results are discussed in section 7. The ablation studies are presented in section 8. The paper is concluded in section 9.

## 2 RELATED WORK

In the literature, various methods have been suggested for LULC mapping [25], [28]. This problem is often approached as pixel-by-pixel segmentation in computer vision [19, 20]. Over the years, many segmentation architectures have been proposed, with U-Net [27] being a popular choice. For instance, Karra et al. [18] trained a large U-Net model using 24,000 (5km x 5km) hand-labeled image chips with ten classes, achieving an 85% accuracy. Though the accuracy is good, some caveats here are the large number of labeled samples required for supervised learning and the coarse resolution of the resulting LULC maps. Bengana and Heikkila [2] tackled the issue of limited labeled samples using a domain adaptation technique called bidirectional learning [21], achieving a mean Intersection over Union (mIoU) of 44.03% with 100m x 100m resolution maps, which is still limited by a coarse resolution. Another study by Hu et al. [15] conducted generalization experiments using continent-wise and season-wise k-fold cross-validation, revealing lower accuracy for out-of-continent or out-of-season data compared to in-continent or in-season data. The above supervised methods for LULC mapping are limited by at least one of the following factors: amount of labeled data, coarser resolution of LULC maps, and lack of scalability and generalizability on different geo-locations.

In recent times, generative AI and foundation models (FM) that leverage their pre-training on diverse multi-modal data to tackle a spectrum of applications have emerged as solutions to address some of the above challenges. Vision Transformer (ViT) [9] has been pre-trained on natural images and has been adapted to several downstream tasks on ImageNet. Note that we adapt this for LULC mapping and is one of the baselines in this paper. SatMAE [5], a ViT based on Masked Auto-Encoder (MAE) [13] has been trained on Sentinel-2 (S2) data at 10m resolution. In contrast to SatMAE which has been pre-trained only on one satellite, S2, in this paper, we focus on Prithvi [16] which is a geospatial FM pre-trained on a combination of Landsat and S2. Prithvi has been pre-trained on Harmonized Landsat Sentinel (HLS) data for the US region and we demonstrate its adaptability for LULC mapping.

In the next section, we will provide an overview of the Prithvi model.

## 3 OVERVIEW OF PRITHVI AND ITS ADAPTATION FOR LULC

Prithvi employs a self-supervised encoder, designed with a ViT [9] architecture and driven by a MAE [14] learning strategy facilitated by a Mean Squared Error (MSE) loss function. This approach incorporates spatial attention spanning multiple patches, along with temporal attention tailored to individual patches. The model processes NASA's HLS data [4] in a video format denoted as (B, C, T, H, W), where B represents the batch size, C signifies the number of channels, T is dedicated to the pivotal temporal dimension crucial for capturing land cover class dynamics, while H and W correspond to height and width dimensions, enabling detailed spatial analysis. By virtue of its design, Prithvi requires substantially lower amount of data for fine-tuning to generate LULC maps, employs dual satellite data (HLS) and generalizes for the entire US region.

We perform various experiments to answer the following hypotheses for Prithvi. The hypotheses are that

- Pre-trained weights encapsulate the learning of features of remote sensed data and are the weights for the base model for downstream tasks.
- Fine-tuning of the pre-trained weights improves the performance of the base model for a given downstream task.

Additionally, we showcase the graceful performance degradation of the fine-tuned model as the fine-tuning dataset size decreases. These experiments were conducted for Prithvi to achieve LULC mapping for the entire US region. The performance is bench marked against U-Net model and ViT models.

## 4 DATA

Remote sensing through satellite imagery is an effective way of monitoring land cover maps due to its scalability across different regions and easy accessibility. The only major disadvantage is the cloud cover as most sensors are unable to obtain data when there are clouds. We have used HLS [3] with a spatial resolution of 30m as input and S2 LULC [18] as the labels for all the experiments in this paper. This dataset is curated by selecting random tiles from the US region for the year 2018 followed by a heuristic for undersampling the majority classes. The HLS-2 (inputs), S2 LULC (labels), and the sampling strategy is described in detail below.

### 4.1 HLS-2 Data

The images from a harmonized surface reflectance product called HLS-2 (version 2.0) data are downloaded from the NASA's land processes distributed active archive center cumulus cloud as cloud optimized GeoTIFFs. The HLS-2 data has been pre-processed to generate and select $224 \times 224$ pixel$^2$ cloud-free patches. We used the blue, green, red, narrow near-infrared (NIR), short wave infrared (SWIR) 1, and SWIR 2 channels. Each HLS tile has an associated cloud mask file, which contains information regarding clouds, cloud shadows, and adjacent to clouds/cloud shadows per pixel [4]. This information is used to find out the regions from the tiles that are cloud-free. These processed cloud-free patches were utilized as an input data for both pre-training and fine-tuning processes.

|  | Water | Trees | Flooded Veg. | Crops | Built area | Bare ground | Snow | Clouds | Rangeland | Mean IoU |
|---|---|---|---|---|---|---|---|---|---|---|
| **Baselines** |  |  |  |  |  |  |  |  |  |  |
| U-Net | 87.5 | 68.56 | 25.36 | 16.51 | 28.49 | 56.40 | 0 | 0 | 44.45 | 36.36 |
| ViT-base | 77.49 | 64.02 | 35.70 | 47.48 | 38.93 | 56.07 | 28.97 | 21.86 | 50.48 | 46.8 |
| **Prithvi** |  |  |  |  |  |  |  |  |  |  |
| Pre-trained | 87.61 | 66.76 | 22.81 | 36.5 | 32.44 | 55.92 | 3.79 | 0.0 | 43.48 | 38.81 |
| **Fine-tuned** | **92.73** | **79.65** | **53.75** | **61.08** | **54.73** | **70.49** | **35.77** | **46.46** | **66.69** | **62.37** |

Table 1: Per class IoU and mean IoU performance of Prithvi model compared to state-of-the-art models, U-Net and ViT-base. Results are evaluated on test data from US region. In "Prithvi Pretrained" we initialize the model with pre-trained weigths, and only learn the decoder weights, keeping the encoder weights frozen. In "Prithvi Finetuned" we initialize the model with pre-trained weights and fine-tune both encoder and decoder.

## 4.2 S2 LULC Data

S2 LULC label data [18] is used as ground truth to fine-tune the Prithvi model. This data has a native spatial resolution of 10m × 10m. S2 LULC label data classifies every 10m × 10m patch to one of the following 10 different classes: No data, Water, Trees, Flooded Vegetations, Crops, Built Area, Bare ground, Snow/Ice, Clouds, and Rangeland.

Label data is downloaded from the S2 10m Land Use/Land Cover dataset provided by Esri (Sources: Esri, Microsoft, Impact Observatory). This data is in Universal Transverse Mercator (UTM) projection and each GeoTiff file is a composite of land cover predictions for each year. The large UTM-sized label tiles are clipped to match the HLS-sized input tiles. Re-gridding and change of resolution are carried out to align the label and input data. Finally, S2 LULC label data is upscaled to 30mx30m spatial resolution to match the resolution of HLS-2 input image patches.

## 5 MODEL

We approached the LULC mapping task as a pixel-by-pixel segmentation task. We utilized the weights from Prithvi's pre-trained encoder model and trained a decoder head on top of it for segmentation. The following sections provide details on the pre-trained model and fine-tuning.

## 5.1 Prithvi Pre-trained Model

We used the pre-trained Prithvi model based on the MAE approach, a successful self-supervised learning method widely used and extended for different data types, including videos [26] and multispectral images [6]. The MAE reconstructs masked images using an asymmetric encoder-decoder architecture. The input image is divided into non-overlapping patches of the same size, and a subset of the patches is randomly masked. The encoder receives only the unmasked patches generating their latent representation. The decoder then receives the latent and masked tokens to perform the image reconstruction task [14].

## 5.2 Fine-tuning

We fine-tune the pre-trained encoder and learn decoder weights for LULC segmentation task. The semantic segmentation decoder used for our downstream task is a Fully Convolutional Network

(FCN) as described in [22]. We used two consecutive FCNs with one convolutional layer in each FCN. The decoder network upscales the embedding that is output by the encoder to the image size and finally maps the output of the projection component to the predicted classes. We have used the same decoder head in ViT-base and Prithvi for a fair comparison.

## 6 EXPERIMENTATION

In this section we talk about the experimental details for demonstrating the performance of Prithvi fine-tuned model.

## 6.1 Training and Test Data

We used 9 HLS tiles[1] from the US region for all our experiments. HLS tiles are $3660 \times 3660$ pixel$^2$ in dimension. We processed the HLS-2 and S2 LULC label data as mentioned in sections 4.1 and 4.2. From these 9 tiles, we generated $224 \times 224$ pixel$^2$ cloud-free patches. We then use a class balancing heuristic described below.

**Handling Class Imbalance**: As S2 LULC label data is naturally imbalanced, steps were then taken to obtain a balanced training dataset. Creating a balanced dataset for pixel-wise segmentation poses more challenges compared to a classification task. The heuristic that we used to create a balanced dataset is explained here- We scan through a large amount of randomly selected $224 \times 224$ pixel$^2$ patches and count the number of pixels in the patches that are assigned to each of the classes. If the number of pixels of a particular class in a particular image is X% or more, then the image is called a sample of that particular class. Like this, we try to get an equal number of samples for every class. For some classes like flooded vegetation, bare ground, clouds, and snow, we did not find images that have X% or more pixels for these images, so we further decreased the value of X to get more data points for these classes.

After class-balancing, we split the data into 396 training, 13 validation, and 156 test patches, each with a good representation of each of the nine classes (except the No data class).

## 6.2 Model Setting and Hyperparameters

In the base setting, we fine-tune the Prithvi pre-trained model for 100 epochs, with a batch size of 6, a learning rate of $6e^{-4}$, and cross-entropy loss. We utilize an AdamW [23] optimizer with a weight

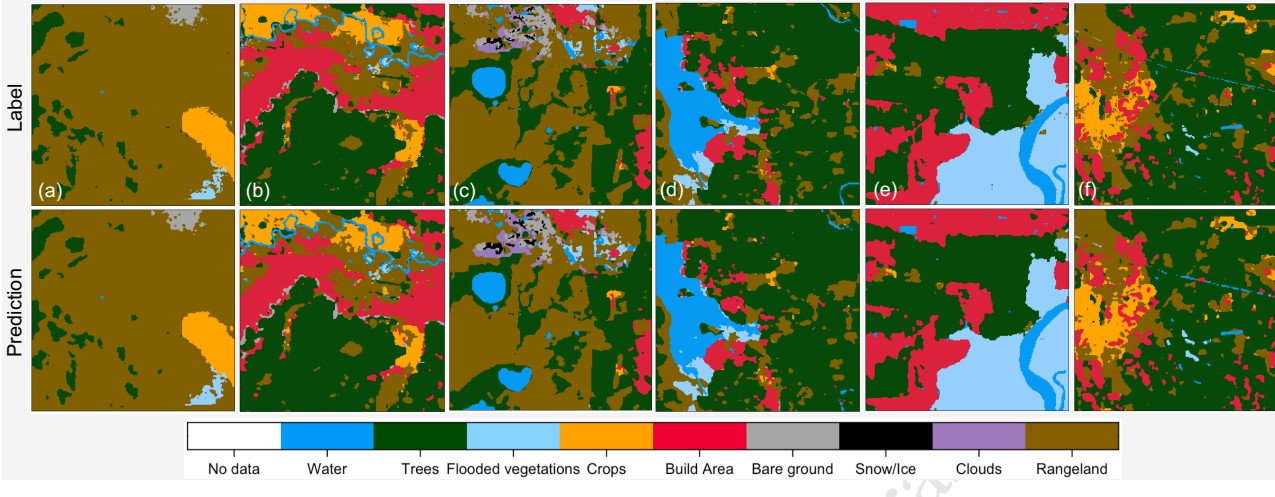

**Figure 1: LULC labels and predictions from Prithvi's fine-tuned model for six sample test patches, representing several classes, are shown here. The predictions demonstrate good agreement with the labels.**

decay of 0.05. The optimal values of the above hyper-parameters were selected after trying several combinations of them.

Our fine-tuning module is currently based on Pytorch via an enhanced version of the MMSegmenation [7] library to deal especially with spatio-temporal data. It supports semantic segmentation tasks (i.e., pixel-level classification).

## 6.3 Baselines

We compared the performance of Prithvi fine-tuned model with a traditional deep-learning model U-Net as well as with ViT-a state-of-the-art foundation model proposed for natural images.

- **U-Net**: U-Net [24] is a fully convolutional symmetric encoder-decoder architecture. We have used the implementation of U-Net which is available in the MMSegmentation package.
- **ViT**: ViT is a transformer-based asymmetric encoder-decoder architecture. ViT [9] splits an image into fixed-size patches, linearly embeds each of them, adds position embeddings, and feeds the resulting sequence of vectors to a standard Transformer encoder, followed by a decoder head.

## 7 RESULTS AND DISCUSSION

### 7.1 Performance Metric

We use the fine-tuned Prithvi model and baseline models described in Section 6.3 for inferencing upon the 156 test patches. The output are patches with LULC classes for each of the pixels in the test patches. In order to evaluate and compare the performance of the models, we use Intersection over Union (IoU) as the performance metrics. IoU is a widely used evaluation metric for image segmentation models. It measures the overlap between the predicted segmentation and the ground truth masks and is calculated as the ratio of intersection area and union area of these 2 masks. We compute IoU for each class using equation 1.

$$IoU = \frac{TP}{TP + FP + FN} \qquad (1)$$

where TP, FP and FN represent number of True Positive, False Positive and False Negative pixels respectively for the class. The IoU metric provides a simple and intuitive way to assess the overlap between the predicted and actual regions with values ranging from 0 (no overlap) to 1 (perfect alignment). We use IoU for measuring the accuracy of the predicted regions against the ground-truth regions for each of the 9 classes and also evaluate the mean IoU (mIoU) by taking the mean of the IoU values for the 9 classes.

### 7.2 Comparison of Prithvi with Baseline

Table 1 shows IoU result for each of the classes in the dataset along with the mIoU for different models. We observe that Prithvi achieved a mIoU of 62.37% and outperforms the baseline models U-Net and ViT, which achieved mIoUs of 36.36% and 46.8% respectively. Prithvi is also able to perform better than the baseline for each of the classes, which is evident from higher IoU per class for Prithvi.

We found that the model could easily detect water, trees, bare ground, and rangeland. However, it struggled with categorizing crops, flooded vegetation, built areas, snow, and clouds and has low IoU for these classes. It is difficult to predict crops and flooded vegetation because of insufficient temporal resolution of the labeled data, as the cropping pattern changes frequently, but the S2 LULC labeled data only has one map per year for a particular region.

In Figure 1, we present six examples of labeled and predicted maps generated using the best-performing fine-tuned model for test images, illustrating a variety of landscapes. A detailed examination reveals that our model excels at predicting the overall structure of these landscapes. However, a closer inspection also highlights its limitations in accurately capturing smaller segments and minor details. Specifically, in the top part of Figure 1(e), the model faces challenges in recognizing some smaller water bodies situated on top of built areas. Similarly, in the top-right and bottom-right sections of Figure 1(d), our model struggles to make precise predictions, notably failing to correctly identify the small blue curves associated with the water class. Examining Figures 1(a), (b), (c), (d), (e), and

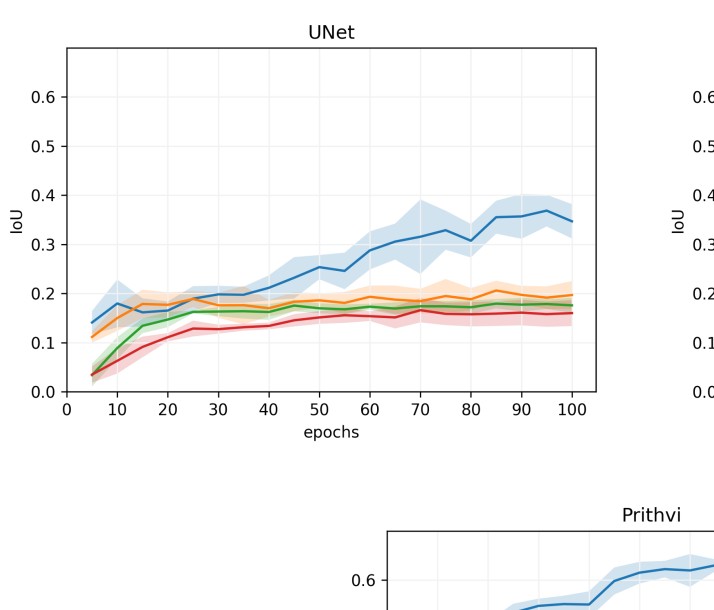
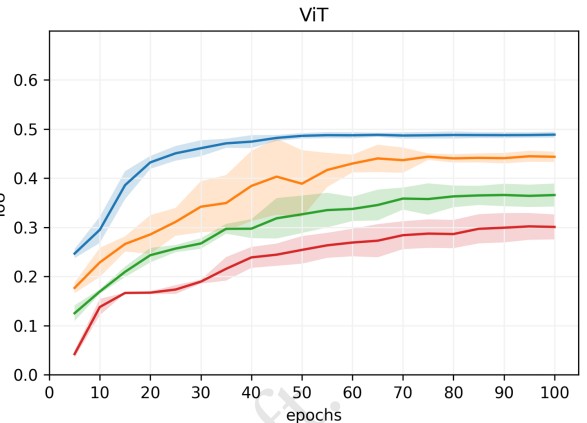
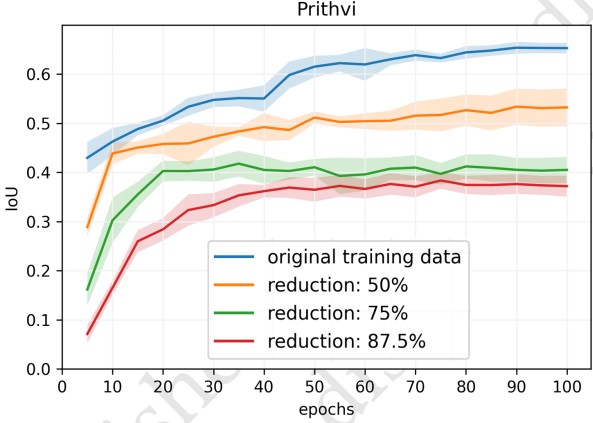

**Figure 2: Data efficiency of UNet, ViT, and Prithvi fine-tuned in terms of reduction of labeled images for fine-tuning the model. Confidence bands represent the standard deviation across 5 different seeds. Each seed corresponds to a different combination of initial random weights. The labels shown in the plot for Prithvi are consistent for UNet and ViT (legends across subplots are identical, as shown in the inset of Prithvi subplot.)**

(f) collectively emphasizes a consistent pattern: the model tends to either underestimate these smaller segments or completely overlook them. This observation implies a potential area for refinement, where enhancing the model's capacity to predict and account for finer details becomes crucial. Understanding these nuances in the model's performance is essential for further developments and improvements. By addressing the model's limitations in capturing smaller-scale features, we can work towards a more accurate and comprehensive representation of varied landscapes in our geospatial predictions.

## 8 ABLATION STUDIES

Ablation studies provide valuable insights into the data requirements and inner workings of fine-tuning models in the LULC mapping, guiding efforts to enhance their performance and robustness. In this paper, we conducted ablation studies by varying (a) the configuration of the pre-trained model and (b) reducing label data.

### 8.1 Configurations of Prithvi

We use the following two configurations of Prithvi model.

(1) Prithvi (Pre-trained)- During segmentation learning, we only update the decoder weights and leave the encoder weights unchanged, which were initialized from a pre-trained model.

(2) Prithvi (Pre-trained finetuned) - Here, we initialize the weights of the encoder from the pre-trained model and also fine-tune them while learning to segment LULC maps.

The mIoU increases from 46.8% to 62.37% as we go from Prithvi (Pre-trained) to Prithvi (Fine-tuned). This clearly demonstrates the effectiveness of fine-tuning the pre-trained weights in improving performance.

### 8.2 Label Reduction Experiments

We have run experiments to understand the impact of input data size on the performance of the fine-tuned geospatial FM. We use the same hyper-parameters as used for the full fine-tuning dataset.

Figure 2 summarizes the experiments carried out. The darker lines show the mean from 5 different random seeds and the lighter bands show the region from $-\sigma$ to $+\sigma$. In these experiments, we have reduced the size of training data by keeping the validation and test set constant. We saw that as we decrease the size of training data, the mIoU also decreases. The performance of the Prithvi model trained on a reduced amount of data is relatively lower but satisfactory. Even with an 87.5% reduction in data, we can still attain a mIoU of 37%.

We have repeated the above data reduction experiments with ViT and UNet baselines. We noticed that for all the data reduction experiments, Prithvi consistently performed better. In Figure 2 we have plotted the IoUs for all these data-reduction experiments with respect to the epochs. We also noticed that the Prithvi model is able to pick up the learning very well in the starting few epochs as compared to UNet and ViT. The UNet's performance drops drastically as we reduce to data set size from 100% to 50%. This set of experiments shows that with limited labeled samples, pre-trained models have an advantage.

## 9 CONCLUSIONS AND FUTURE WORK

Accurate quantification of land cover and its spatio-temporal evolution is essential to achieve several sustainable development goals adopted by the United Nation member states for the social, economic and environmental development of our planet and its inhabitants. In this paper, we have addressed some of the challenges in the current state-of-the-art LULC mapping methods, particularly the scarcity of labeled samples for training and the difficulty in generalizing across different land classes. We evaluated the performance of the Prithvi model and compared it to other available state-of-the-art models. The results look very promising both in terms of the IoU metric and the prediction maps. The pre-trained weights learned from a large corpus of Geospatial data have proved to be very useful, especially with limited labeled samples available. Experiments with global data for fine-tuning have also been initiated on our end. Here are some preliminary results: we have achieved an aggregate mIoU of 62.37%. While this result is encouraging, there is still room for further refinement. We are considering the following as future work. We are working on a rigorous sampling strategy to achieve class balance in the global dataset for fine-tuning. We are also planning to use a global pre-trained model instead of a US-based pre-trained model. Additionally, we are exploring any known technical aspects that could enhance model performance on a global scale. We are also expanding the fine-tuning procedure to incorporate data from surveys (such as GLanCE [11], NASA) that are sparsely distributed across different geographies.

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

Received 28 May 2024

