# OpenReview forum: "Finetuning the Geospatial Foundation model for Land Cover mapping"
_KDD.org/2024/Workshop/Fragile_Earth — Fragile Earth FullPresentation_

### Official Review · Reviewer_Q5oN · 2024-07-12
**Well structured paper that describes a fine tuning strategy for GFM**

**Rating:** 8
**Confidence:** 4

**Review:**

The paper explores opportunities to fine tune GFM, more specifically Prithvi, to use the model for land use/cover classification problems.  The authors explore the idea of using few-shot model tuning to reduce annotations as well as model development needs for specific LULC cases. The paper uses HLS-2 imagery for both fine tuning and inference purposes. Compared to the existing methods, the proposed approach looks to perform well. Despite Table 1 shows that the proposed model is substantially better than specialized models, i.e. U-Net, I believe the performance gap may be due to the U-Net model was not being trained. Therefore, I take the results with a skeptism. I believe authors should have described a little bit more details in terms of how those baseline models were being used. I believe the paper provides good insights on the applicability of large foundational models on specialized use cases. Thus, I will be happy to see it in our workshop.

---

### Official Review · Reviewer_nRuR · 2024-07-13
**A good paper addressing a critical problem of LULC with impressive results**

**Rating:** 8
**Confidence:** 4

**Review:**

This paper addresses the problem of critical importance to sustainable goals, namely, that of Land use and land cover (LULC) mapping, by fine-tuning of a pre-trained geospatial foundation model (called "Prithvi") and demonstrate its superiority compared to the state-of-the-art approaches, U-Net and ViT. The results show that Prithvi achieves considerable improvement over the baselines. The paper is well-motivated and shows promising results for an important problem for sustainability. Having it in the workshop will benefit the audience.
As a comment, it would be beneficial to have a more detailed description of Prithvi since it plays a central role to the paper's thesis.

---

### Decision · Program_Chairs · 2024-07-24

Accept (Full Presentation)